# A Metalloproteinase Induces an Inflammatory Response in Preadipocytes with the Activation of COX Signalling Pathways and Participation of Endogenous Phospholipases A_2_

**DOI:** 10.3390/biom11070921

**Published:** 2021-06-22

**Authors:** Priscila Motta Janovits, Elbio Leiguez, Viviane Portas, Catarina Teixeira

**Affiliations:** 1Laboratório de Farmacologia, Instituto Butantan, São Paulo 05503-900, Brazil; elbio.leiguez@butantan.gov.br; 2Centre of Excellence in New Target Discovery (CENTD), Instituto Butantan, São Paulo 05503-900, Brazil; viviane.portas@butantan.gov.br; 3Laboratório de Desenvolvimento e Inovação, Instituto Butantan, São Paulo 05503-900, Brazil

**Keywords:** metalloproteinase, preadipocytes, prostaglandins, adipokines, cytokines

## Abstract

Matrix metalloproteinases (MMPs) are proteolytic enzymes that have been associated with the pathogenesis of inflammatory diseases and obesity. Adipose tissue in turn is an active endocrine organ capable of secreting a range of proinflammatory mediators with autocrine and paracrine properties, which contribute to the inflammation of adipose tissue and adjacent tissues. However, the potential inflammatory effects of MMPs in adipose tissue cells are still unknown. This study investigates the effects of BmooMPα-I, a single-domain snake venom metalloproteinase (SVMP), in activating an inflammatory response by 3T3-L1 preadipocytes in culture, focusing on prostaglandins (PGs), cytokines, and adipocytokines biosynthesis and mechanisms involved in prostaglandin E_2_ (PGE_2_) release. The results show that BmooMPα-I induced the release of PGE_2_, prostaglandin I_2_ (PGI_2_), monocyte chemoattractant protein-1 (MCP-1), and adiponectin by preadipocytes. BmooMPα-I-induced PGE_2_ biosynthesis was dependent on group-IIA-secreted phospholipase A_2_ (sPLA_2_-IIA), cytosolic phospholipase A_2_-α (cPLA_2_-α), and cyclooxygenase (COX)-1 and -2 pathways. Moreover, BmooMPα-I upregulated COX-2 protein expression but not microsomal prostaglandin E synthase-1 (mPGES-1) expression. In addition, we demonstrate that the enzymatic activity of BmooMPα-I is essential for the activation of prostanoid synthesis pathways in preadipocytes. These data highlight preadipocytes as important targets for metalloproteinases and provide new insights into the contribution of these enzymes to the inflammation of adipose tissue and tissues adjacent to it.

## 1. Introduction

MMPs are a family of zinc-dependent endoproteases responsible for the degradation of various proteins of the extracellular matrix (ECM). In physiological conditions, when the expression and activity of MMPs are under strict control, these enzymes play important roles in tissue remodelling, host defense, angiogenesis, and immune response, as well as cell proliferation, migration, and differentiation [1,2,3,4]. However, when there is an imbalance between the levels of activated MMPs and their tissue inhibitors, these enzymes become important in the pathogenesis of several inflammatory diseases, such as arthritis, atherosclerosis, and obesity [1,3,5,6,7,8,9]. Although the precise mechanisms of action of MMPs have not been completely described, these enzymes have been shown to trigger inflammatory reactions at various levels, regulating the recruitment of inflammatory cells to the site of inflammation by the processing of ECM components, growth factors, cytokines, and chemokines [1,10,11,12]. Accordingly, increased circulating plasma levels of MMPs and increased expression of these enzymes, mainly of MMP-3, MMP-9, and MMP-13, have been reported in the inflamed tissues of patients suffering from inflammatory diseases, such as rheumatoid arthritis, osteoarthritis, bowel disease, and neuroinflammation [1,3,8,10]. Presently, MMPs have been associated with the development of obesity by promoting adipocyte differentiation, as well as adipose tissue remodelling [5,13,14,15,16]. In this context, levels of MMP-2, MMP-8, and MMP-9 were found to be increased in the adipose tissue of obese patients [5,6,13]. However, despite the roles of MMPs in obesity development, the actions of these enzymes on adipose tissue cells are still unknown.

Adipose tissue is known to secrete a range of proinflammatory mediators with autocrine and paracrine properties, such as the adipose-tissue-derived adipokines leptin, resistin, and adiponectin; cytokines; chemokines; and PGs, which contribute to the development and progression of inflammatory diseases [17,18,19,20,21,22,23]. Among the mediators produced by the adipose tissue, PGE_2_, which is enzymatically converted from arachidonic acid (AA) by the COXs and terminal PGE synthases (PGES), is the predominant PG produced by the adipose tissue and plays a pivotal role in the pathogenesis of several inflammatory diseases [20,24,25]. In inflammatory conditions, PGE_2_ is known to mediate vasodilation, vascular permeability, and pain [26,27]. Regarding obesity, PGE_2_ has been implicated in the regulation of adipose tissue functions and the development of obesity. In this regard, PGE_2_ and COX-2 levels have been found to be increased in the adipose tissue of obese patients [20,28,29,30,31]. Despite modulatory effects in this tissue by the suppression of leukocyte inflammatory activity, several pieces of evidence have shown that PGE_2_ exerts antilipolytic actions that lead to increased adipose tissue mass, a factor that has been associated with increased risk of diabetes and cardiovascular disease [20,31,32,33,34].

Adipose tissue is largely composed of two fractions, mature adipocytes and the stromal vascular fraction (SVF), which is responsible for the generation of many of the proinflammatory mediators secreted by this tissue [35]. Within the SVF, preadipocytes, the undifferentiated precursors of mature adipocytes, account for up to 50% of the cells in human adipose tissue, and when compared to mature adipocytes, have a propensity for greater inflammatory response through the activation of nuclear factor-κB (NF-κB) and mitogen-activated protein kinase (MAPK) signalling [35,36,37]. In the adipose tissue, MMPs are known to participate in the differentiation of preadipocytes into mature adipocytes and in the development of this tissue by promoting ECM remodelling [7,38,39,40]. However, the inflammatory actions of MMPs in adipose tissue cells and mechanisms leading to the production of PGE_2_ in these cells are still unknown.

SVMPs share structural and functional homology with mammalian MMPs and have been grouped within the M12 family of metalloproteinases, which belong to the metzincin superfamily of these proteinases [41,42]. This superfamily is characterised by the presence of a consensus zinc-binding sequence (HEXXHXXGXXH) followed by a conserved loop, Met-turn, in the catalytic domain [1,2,42,43]. The SVMPs have been classified into three main classes (P-I to P-III) based on their domain structure and size. Notably, P-I class SVMPs present only the catalytic metalloproteinase domain, and as with MMPs, are able to degrade ECM components, activate inflammatory cells, and induce inflammatory events in several experimental models. P-I SVMPs are, therefore, useful tools for studies of the biological effects of MMPs, including inflammatory disease development [44,45,46].

Based on this, we investigated the ability of BmooMPα-I, a P-I class metalloproteinase isolated from *Bothrops moojeni* snake venom, to activate the inflammatory response by preadipocytes in culture with a focus on (i) the release of PGE_2_ and PGI_2_, cytokines, and adipokines; and (ii) the mechanisms involved in the release of PGE_2_ induced by this metalloproteinase. For this purpose, we evaluated the participation of COX-1, COX-2, mPGES-1, endogenous PLA_2_s, and metalloproteinase catalytic activity in the BmooMPα-I-induced release of PGE_2_. Our results are the first to demonstrate the ability of a metalloproteinase to activate preadipocytes for production of PGE_2_, PGI_2_, monocyte chemoattractant protein 1 (MCP-1), and adiponectin. The BmooMPα-I-induced production of PGE_2_ is dependent on the activation of endogenous cPLA_2_-α and sPLA_2_-IIA and the COX-1/COX-2/mPGES-1 pathway. The catalytic activity of BmooMPα-I is critical for the activation of PG biosynthesis pathways.

## 2. Materials and Methods

### 2.1. Chemicals and Reagents

MTT and L-glutamine were obtained from USB (Cleveland, OH, USA). Dulbecco’s modified Eagle’s medium (DMEM) and fetal bovine serum (FBS) were purchased from Life Technologies (São Paulo, SP, Brazil). Gentamicin was purchased from Schering-Plough (Whitehouse Station, NJ, USA) and DMSO was purchased from Amresco (Solon, OH, USA). Mouse mAb anti-β-actin was purchased from Sigma-Aldrich (St. Louis, MO, USA) and polyclonal antibodies against COX-1, COX-2 and mPGES-1 were purchased from Cayman Chemical Company (Ann Arbor, MI, USA). PGE_2_; PGI_2_ enzyme immunoassay kit; and the compounds SC-560 NS-398, pyrrolidine-2 (PYR-2), FKGK11, KH064, and Batimastat (BB-94) were also purchased from Cayman Chemical Company (Ann Arbor, MI, USA). Secondary antimouse and antirabbit antibodies conjugated to HRP and nitrocellulose membrane were obtained from GE Healthcare (Buckinghamshire, UK), while the leptin, resistin, and adiponectin immunoassay kit was purchased from Thermo Fisher Scientific (Waltham, MA, USA). The cytometric bead assay (CBA) kit was purchased from BD Bioscience (San Jose, CA, USA).

### 2.2. BmooMPa-I

The metalloproteinase BmooMPα-I, isolated from *Bothrops moojeni* venom, was purified by ion-exchange chromatography on a Hitrap^®^ DEAE Fast Flow column. Gel filtration was performed on a Tosoh G2000SWxL column and affinity chromatography was performed on a Hitrap^®^ Heparin High Performance column [47]. All chromatographic steps were performed using liquid chromatography on AKTA AVANT from GE Healthcare (Buckinghamshire, UK). The homogeneity of the enzyme was determined by sodium dodecyl sulphate–polyacrylamide gel electrophoresis (SDS-PAGE) under reducing conditions and confirmed by mass spectrometry. The results obtained via mass spectrometry demonstrated a single protein peak with a molecular weight of 24.5 kDa compatible with the metalloproteinase BmooMα-I. The proteolytic activity of BmooMPα-I was confirmed using the fluorescence resonance energy transfer substrate (FRET Abz-FRSSRQ-EDDnp), which contains a universal sequence recognized by different classes of proteases [48]. The absence of endotoxin contamination in the BmooMPa-I batches used was demonstrated by the quantitative limulus amebocyte lysate (LAL) test [49], which revealed undetectable levels of endotoxin (<0.125 EU/mL). The enzyme was lyophilised, stored at −20 °C, and dissolved in DMEM medium just before use.

### 2.3. Cytotoxicity Assay

The cytotoxicity of BmooMPα-I toward the 3T3-L1 preadipocyte was evaluated using the MTT assay as previously described [50]. In brief, 4 × 10^3^ preadipocytes per well were plated in 96-well plates in DMEM, supplemented with 40 μg/mL gentamicin sulphate and 2 mM L-glutamine, then incubated with BmooMPα-I (0.24 µM), COX, or PLA_2_ inhibitors or Batimastat diluted in medium or with the same volume of medium alone (control) for 1, 3, 6, 12, 24, and 48 h in a humidified atmosphere (5% CO_2_) at 37 °C. MTT (5 mg/mL) was dissolved in PBS and filtered for the removal of insoluble residues and sterilisation. MTT stock solution (10% in culture medium) was added to all wells in each assay and plates were incubated for 3 h at 37 °C. Dimethyl sulfoxide (DMSO) (100 μL) was added to wells and mixed thoroughly for 30 min at room temperature. Absorbances were then recorded at 540 nm in a microtiter plate reader. The results were expressed as percentages of viable cells, considering control cells incubated with medium alone as 100% viable.

### 2.4. 3T3-L1 Cell Culture and Stimulation

The 3T3-L1 preadipocytes obtained from the American Type Culture Collection were cultured as described in a previous study [35]. Briefly, 5 × 10^3^ preadipocytes per well were seeded in 12-well culture plates and maintained in culture medium for 48 h before stimulation, according to the experimental protocol. Preadipocytes were serum-starved in DMEM containing 1% (*v*/*v*) gentamicin sulphate supplemented with 1% (*v*/*v*) L-glutamine for 18 h prior to all treatments. Cell homogenates were collected and used for the Western blotting analysis of COX-1, COX-2, and mPGES-1 protein expression, and supernatants of each treatment were used to measure lipid mediators PGE_2_ and PGI_2_ and the adipokines leptin, resistin, and adiponectin by enzyme immunoassay (EIA), as well as the cytokines MCP-1, TNF-α, IL-1β, KC, IL-6, and IL-10 by cytometric bead array (CBA). Cells were stimulated with BmooMPα-I (0.24 µM), diluted in DMEM (2.5% FBS) or DMEM alone (control) for selected periods, and maintained in a humidified atmosphere (5% CO_2_) at 37 °C. To investigate the mechanism involved in the PGE_2_ biosynthesis and the participation of the metalloproteinase enzymatic activity in the effects induced by BmooMPα-I, selective inhibitors were used at previously tested concentrations: 1 μM SC-560 (COX-1 inhibitor) [51] and NS-398 (COX-2 inhibitor) [52], 1 µM PYR-2 (cPLA_2_-α inhibitor) [53], 1 µM FKGK11 (calcium-independent phospholipase A_2_ (iPLA_2_) inhibitor) [54], 10 µM KH064 (sPLA_2_-IIA inhibitor) [55], and 32 µM Batimastat (metalloproteinase inhibitor) [56]. All stock solutions were prepared in DMSO and stored at −20 °C. Aliquots were diluted in DMEM immediately before use. The concentration of DMSO was always lower than 1%. The viability of cells treated with inhibitors was evaluated by MTT assay. No significant changes in cell viability were registered with any of the tested agents or the vehicle at the concentrations used.

### 2.5. Inhibition of Metalloproteinase Activity

BmooMPα-I was incubated with the peptidomimetic hydroxamate metalloproteinase inhibitor Batimastat (32 µM) at 37 °C for 30 min prior to incubation with cells [56]. The 3T3-L1 preadipocytes plated in 12-well plates were then incubated for 6 h at 37 °C with aliquots of BmooMP*α*-I (0.24 µM) inhibited by Batimastat and controls, which included BmooMP*α*-I without Batimastat, Batimastat alone, or vehicle (DMSO). Cell homogenates were collected for the evaluation of COX-2 protein expression and supernatants were collected for the quantification of PGE_2_.

### 2.6. Western Blotting

The protein expression of COX-1, COX-2, and mPGES-1 from cell homogenates was detected by Western blotting. Briefly, BmooMPα-I-stimulated and non-stimulated cells were lysed with 100 mL of a sample buffer (0.5 M Tris HCl, pH 6.8, 20% SDS, 1% glycerol, 1 M β-mercaptoethanol, 0.1% bromophenol blue) and boiled for 10 min. Samples were resolved by SDS-PAGE on 10% bis-acrylamide gels overlaid with a 5% stacking gel. Proteins were transferred to nitrocellulose membranes using a Mini Trans-Blot system (Bio-Rad Laboratories, Richmond, CA, USA). Membranes were then blocked for 1 h with 5% albumin in Tris-buffered saline (20 mM Tris, 100 mM NaCl and 0.5% Tween 20, pH 7.2) and incubated overnight at room temperature with primary antibodies against COX-1, COX-2, and mPGES-1 (1:500 dilution) or for 1 h with the primary antibody against β-actin (1:3000 dilution). Membranes were washed and incubated with the appropriate secondary antibody conjugated to horseradish peroxidase. Immunoreactive bands were detected by the entry-level peroxidase substrate for enhanced chemiluminescence, according to the instructions of the manufacturer (GE Healthcare). Band densities were quantified with an ImageQuant LAS 4000 mini densitometer (GE Healthcare) using the image analysis software ImageQuant TL (GE Healthcare).

### 2.7. Prostanoid, Cytokine and Adipokine Quantification

PGE_2_ and PGI_2_, as well as the adipokines leptin, resistin, and adiponectin, were measured using EIA kits, while cytokines (MCP-1, TNF-α, IL-1β, KC, IL-6, IL-10) were quantified using a CBA kit from supernatants of preadipocytes incubated with each treatment. Kits were used following the instructions of the manufacturer.

### 2.8. Statistical Analysis

Data are expressed as means ± SEM (*n* = 3–4). Multiple comparisons among groups were performed using ANOVA, and as a post-test, the Bonferroni test. Differences between experimental groups were considered significant for *p*-values < 0.05. All statistical tests were performed using GraphPad Prism version 5 software (GraphPad, San Diego, CA, USA).

## 3. Results

### 3.1. Metalloproteinase BmooMPα-I Induces the Release of PGE_2_ and PGI_2_ by Cultured Preadipocytes

PGE_2_ and PGI_2_ are recognised as key mediators for the pathogenesis of inflammatory diseases [57,58,59,60]. PGE_2_ has been reported to be released by macrophages upon MMP stimulus [61]; however, whether this class of enzymes stimulates preadipocytes to release PGE_2_ and PGI_2_ has yet to be established. To investigate the action of BmooMPα-I in preadipocytes, we first assessed the ability of this metalloproteinase to induce the release of PGE_2_ in cultured preadipocytes by testing three concentrations of BmooMPα-I (0.06, 0.12, and 0.24 μM), which were added to the culture for 24 h. At these concentrations, BmooMPα-I did not affect cell viability after 6 or 24 h of incubation, as assessed by MTT assay. After establishing that BmooMPα-I induced a maximal effect at 0.24 µM (Figure 1A), we evaluated the time course of PGE_2_ and PGI_2_ release induced by BmooMPα-I. To that end, BmooMPα-I (0.24 µM) was added to the culture for 1, 3, 6, 12, and 24 h, and PGE_2_ and PGI_2_ release was evaluated by ELISA. As shown in Figure 1B, BmooMPα-I induced a significant release of PGE_2_ from 3 to 24 h and of PGI_2_ from 12 to 24 h (Figure 1C) when compared with control cells incubated with culture medium alone. These results indicate that BmooMPα-I has the ability to directly stimulate preadipocytes to produce prostanoids.

### 3.2. BmooMPα-I-Induced Release of PGE_2_ Is Dependent on COX-1 and COX-2

COX-1 and COX-2 are key enzymes responsible for the production of prostanoids from AA [62]. COX-2 expression is induced by inflammatory stimuli and has largely been associated with the increased production of PGs in inflammatory conditions [63,64]. In addition, PGE_2_ is one of the most abundant lipid mediators produced in the adipose tissue [25]. Therefore, to investigate the mechanisms underlying PGE_2_ release induced by the metalloproteinase BmooMPα-I in preadipocytes, we evaluated the participation of COX-1 and COX-2 in this effect. Preadipocytes were treated for 1 h with either compound SC-560 or compound NS-398, selective inhibitors of COX-1 and COX-2, respectively, or their vehicle. PGE_2_ release was evaluated after 12 h of incubation with BmooMPα-I. As shown in Figure 2A, preadipocytes preincubated with vehicle followed by stimulation with BmooMPα-I induced significant release of PGE_2_ when compared with basal preadipocytes (negative control). Pretreatment of cells with SC-560 abolished BmooMPα-I-induced PGE_2_ release in comparison with preadipocytes treated with vehicle followed by stimulation with BmooMPα-I (positive control). Pretreatment of cells with NS-398 significantly reduced the release of PGE_2_ induced by BmooMPα-I (Figure 2A) in comparison with the positive control; pretreatment of preadipocytes with the association of both SC-560 and NS-398 abolished this release when compared with the positive control. These results suggest that both COX-1 and COX-2 play an important role in BmooMPα-I-induced PGE_2_ production in preadipocytes. Based on the above results, we next investigated whether BmooMPα-I can induce the protein expression of COX-1 and COX-2 by preadipocytes. As demonstrated in Figure 2B,C, the incubation of cells with BmooMPα-I did not affect the constitutive expression of COX-1 in comparison with the basal control but it significantly increased the protein expression of COX-2 after 3 h and up to 24 h of incubation (Figure 2B,D). Taken together, these findings indicate that the production of PGE_2_ induced by BmooMPα-I depends on the COX-1 and COX-2 signalling pathways and suggest that the ability of BmooMPα-I to upregulate COX-2 expression at the translational level is one of the mechanisms of the increased production of PGE_2_ seen at the later time interval (24 h). 

### 3.3. BmooMPα-I Induces mPGES-1 Protein Expression

Since mPGES-1 is an inducible enzyme responsible for the final step of the PGE_2_ biosynthetic cascade [62], we further explored the mechanisms involved in BmooMPα-I-induced PGE_2_ production in preadipocytes by assessing mPGES-1 protein expression after stimulation with the metalloproteinase. Figure 3A,B shows that the stimulation of cells with BmooMPα-I did not alter mPGES-1 protein expression when compared with control cells, suggesting that this mechanism is not involved in the BmooMPα-I-induced production of PGE_2_.

### 3.4. Metalloproteinase Enzymatic Activity Is Important for BmooMPα-I-Induced Release of PGE_2_ and COX-2 Expression

The catalytic domain is conserved among all classes of the metalloproteinase enzyme superfamily [2]. Therefore, to investigate the importance of the enzymatic activity of metalloproteinases in the effects triggered by BmooMPα-I in preadipocytes, in particular in the release of PGE_2_ and protein expression of COX-2, two fundamental events in inflammation, the catalytic domain of this enzyme was inhibited by incubation with the compound Batimastat. Batimastat is a hydroxamate peptidomimetic that binds specially to zinc ions in MMPs and inhibits enzyme activities [64,65]. As shown in Figure 4, the inhibition of the enzymatic site of BmooMPα-I abolished the release of PGE_2_ (Figure 4A) and COX-2 expression (Figure 4B,C) after 6 h of incubation when compared with the positive control incubated with the active enzyme. This indicates that the enzymatic activity of BmooMPα-I is important for the activation of prostanoid synthesis pathways in preadipocytes.

### 3.5. cPLA_2_-α and sPLA_2_-IIA Participate in the BmooMPα-I-Induced Release of PGE_2_

It is well known that PLA_2_s are upstream enzymes in the signalling cascade involved in the production of eicosanoids, including PGs. Additionally, it is well documented that PLA_2_s hydrolyse membrane phospholipids, resulting in the release of AA, which is further converted into biologically active PGs, such as PGE_2_, by the COX enzymes and PG synthases [66]. Taking this information into account, we investigated the role of distinct PLA_2_s in PGE_2_ release induced by BmooMPα-I. We used pharmacological approaches to identify the critical PLA_2_s involved in this effect of BmooMPα-I. For this purpose, cells were pretreated with effective concentrations of PYR-2 compound, a specific inhibitor of cPLA_2_-α; FKGK11 compound, an inhibitor of iPLA_2_; or KH064 compound, a sPLA_2_-IIA inhibitor. As marked release of PGE_2_ was detected after 12 h of incubation with BmooMPα-I, we evaluated the effects of the pharmacological compounds in the same time interval. As shown in Figure 5, the treatment of cells with PYR-2 or KH064 compounds abolished the stimulatory effect of BmooMPα-I in comparison with the positive control. The treatment of cells with FKGK11 compound, however, did not affect the release of PGE_2_ induced by this metalloproteinase. These findings indicate that cPLA_2_-α and sPLA_2_-IIA but not iPLA_2_ are involved in the production of PGE_2_ induced by BmooMPα-I in preadipocytes. 

### 3.6. BmooMPα-I Induces Release of MCP-1 and Adiponectin by Cultured Preadipocytes

Cytokines and adipokines are important mediators that participate in the development of inflammatory diseases and in adipose tissue inflammation [67,68,69,70,71]. To further investigate the inflammatory effects of MMPs in preadipocytes, we evaluated whether BmooMPα-I can induce the release of cytokines and adipokines by these cells. Thus, preadipocytes were incubated with BmooMPα-I (0.24 µM) for 1, 3, 6, 12, 24, or 48 h, and the release of those mediators was evaluated by ELISA. As shown in Figure 6, BmooMPα-I induced significant release of MCP-1 from 3 to 24 h (Figure 6A) and of adiponectin (about 80% increase) at 48 h of incubation (Figure 6D) when compared with control cells incubated with culture medium alone. This metalloproteinase did not induce significant release of IL-6 (Figure 6B), KC (Figure 6C), leptin (Figure 6E), or resistin (Figure 6F) compared with control cells. The release of TNF-α, IL-1β, and IL-10 was not detectable in our experimental condition. The obtained results show for the first time the capacity of a metalloproteinase to induce release of MCP-1 and adiponectin by preadipocytes and are in line with the reported proinflammatory activity of MMPs and SVMPS, especially BmooMPα-I.

## 4. Discussion

Matrix metalloproteinases have been associated with the pathogenesis of various inflammation-related diseases, ranging from cancer to chronic inflammatory diseases, including obesity [1,3,5,6,7,8,9]. Levels of MMPs are elevated in the adipose tissue of obese patients [6] but the inflammatory effects of MMPs in this tissue are not fully understood. Comprehending the inflammatory action of metalloproteinases on adipose tissue cells, such as preadipocytes, is critical to identify the factors underlying adipose tissue inflammation, which lead to obesity or inflammation of adjacent tissues due to the impacts of inflammatory mediators reaching these tissues [68]. We have demonstrated the ability of BmooMPα-I, a P-I class SVMP, to elicit an inflammatory response in preadipocytes in culture. Our results show that BmooMPα-I induced a marked release of PGE_2_ and PGI_2_ from preadipocytes in culture. The concentrations of BmooMPα-I able to induce release of these prostaglandins were comparable to those described for relevant MMPs, such as MMP-2, in the plasma of obese patients [6]; therefore, it is plausible to suggest that PGE_2_ and PGI_2_ can mediate the inflammatory actions of metalloproteinases in the adipose tissue and exert paracrine actions in structures adjacent to this tissue, contributing to the development of distinct inflammatory conditions, such as arthropathies [18,23]. Although the ability of MMP-1 and MMP-3 to release PGE_2_ from isolated macrophages and of an SVMP, BaP1, to release this mediator by fibroblast-like synoviocytes [61,72] has previously been reported, to our knowledge this is the first report providing evidence of a metalloproteinase inducing the production of PGE_2_ and PGI_2_ by adipose tissue cells.

Prostanoids are produced from the metabolism of AA by COX-1 and COX-2 and are fundamental mediators in inflammatory responses and the perpetuation of inflammation signs and symptoms [60,73]. In the adipose tissue, prostanoids, mainly PGE_2_, are implicated in the process of the differentiation of preadipocytes into mature adipocytes, leading to the increased mass of this tissue [25,28,74,75,76]. Taking into account the marked release of PGE_2_ in preadipocytes stimulated by BmooMPα-I and the importance of this mediator in inflammation, we investigated the mechanisms involved in PGE_2_ biosynthesis induced by BmooMPα-I. Our findings from our pharmacological approach showed that PGE_2_ production induced by BmooMPα-I is dependent on the activation of COX-1 and COX-2 isoforms in preadipocytes. As an additional mechanism, BmooMPα-I upregulated COX-2 protein expression but did not alter the protein levels of the inducible terminal synthase mPGES-1. As expected, BmooMPα-I did not affect protein expression of the constitutive COX-1. These results indicate that adipose tissue cells are targets for the action of metalloproteinases and align with the idea that COX-derived mediators can be second messengers of metalloproteinases for the development of inflammation in these cells. The mechanism by which BmooMPα-I upregulates the expression of COX-2 in preadipocytes has not yet been investigated. However, an autocrine effect of the chemokine MCP-1 may be involved in this effect, since this mediator is released from preadipocytes upon BmooMPα-I stimulus and has been previously reported to lead to COX-2 expression [76,77,78].

PLA_2_s are lipolytic enzymes that act on membrane glycerophospholipids for the release of AA, the main substrate of COX enzymes and the precursor of PGs, including PGE_2_. PLA_2_s have also been demonstrated to be key enzymes in triggering diverse inflammatory diseases [79,80,81,82,83]. In this work, we investigated the participation of these enzymes in the BmooMPα-I-induced release of PGE_2_. Our results, which showed that the pharmacological inhibition of cytosolic PLA_2_s or group-IIA-secreted PLA_2_s markedly reduced the BmooMPα-I-induced release of PGE_2_, indicated that both cytosolic and group-IIA-secreted PLA_2_s are important players in the generation of PGE_2_ following BmooMPα-I stimulus. These findings are evidence of a link between metalloproteinases and PLA_2_s, by which the venom metalloproteinase stimulates preadipocytes to produce PGE_2_; they are in accordance with previous studies demonstrating a link between metalloproteinases and PLA_2_ for the production of PGE_2_ by fibroblast-like synoviocytes [72]. On the other hand, a negative regulation of a cardiac sPLA_2_ by MMP-2 in a proinflammatory setting has been previously reported, since in MMP-2 deficient mice the production and release of a unique sPLA_2_ was increased in cardiomyocytes [84,85]; thus, MMPs may act as modulators of distinct members of the PLA_2_ family and may regulate inflammation signaling, both when expressed in excess and when underexpressed. The mechanisms involved in BmooMPα-I-induced activation of cPLA_2_-α and sPLA_2_-IIA were not addressed in our study; however, in the case of cPLA_2_-α, some mechanisms may be suggested. First, the release of molecular patterns associated with cell damage (DAMPs) via the action of BmooMPα-I on cell membrane may lead to activation of Toll-like receptors (TLR) followed by production of inflammatory mediators, including PGE_2_. In this context, the regulation of cPLA_2_ activation and lipid generation by TLR4 signaling [86] and activation of TLR4 by DAMPs [87] were previously reported. Moreover, production of DAMPS by a snake venom metalloproteinase and MMPs were recently demonstrated [86,88]. Second, the potential activation of PAR receptor by BmooMPα-I may result in increased phosphorylation of cPLA_2_ and the release of AA, since activation of the PAR receptor by proteases leading to activation of cPLA_2_-α coupled to COX-1 has been reported [89].

In addition, our results showing that the inhibition of the BmooMPα-I catalytic domain abolished PGE_2_ release and COX-2 protein expression indicate that the catalytic activity of BmooMPα-I is essential for the activation of the biosynthetic pathway for the production of prostanoids in preadipocytes. Considering that the catalytic domain of metalloproteinases was demonstrated to be responsible for the degradation of ECM components and for the processing of non-matrix proteins, such as cytokines, chemokines, and receptors [11,72], our findings provide insight into the role of the catalytic domain of MMPs in the release of lipid mediators triggered by this class of enzymes.

Extensive research has shown that the inflammation of adipose tissue may be a major factor in the development of metabolic diseases and that cytokines and adipokines released by adipose tissue contribute to the inflammation of adjacent tissues [18,23,76,90]. Our results showing that BmooMPα-I induced a marked release of MCP-1 and a late release of adiponectin from preadipocytes in culture contribute to the information regarding the inflammatory effects of this metalloproteinase on preadipocytes. It is known that MCP-1 initiates adipose tissue inflammation by inducing the recruitment of monocytes to the tissue, contributing to obesity onset [35,71]. Moreover, MCP-1 biosynthesis has been reported in the intra-articular adipose tissue, with a role in the pathogenesis of joint diseases [91]; thus, the release of MCP-1 by preadipocytes under the stimulus of a metalloproteinase may contribute not only to adipose tissue inflammation, but also to the progression of inflammation in adjacent tissues. Moreover, a role of MCP-1 in activation of PLA_2_s induced by BmooMPα-I may be suggested, since MCP-1 has been shown to induce the phosphorylation of cPLA_2_-α and the rapid release of AA by human leukocytes [92,93]. Adiponectin is mainly produced by the adipose tissue and has been recognised as an important modulator of the immune system [69]. Unlike most other adipokines, adiponectin appears to have a protective role in metabolic syndrome and diabetes mellitus type 2 [69,70]. However, in inflammatory joint diseases, this adipokine has been highlighted as an important player in synovitis and joint destruction, as it acts as a proinflammatory mediator, inducing the production of MCP-1, PGE_2_, and MMPs and the expression of COX-2 by synovial cells [23,69,77]; thus, the release of adiponectin by preadipocytes under the stimulus of a metalloproteinase provides insight into the participation of MMPs and adipose tissue cells in joint inflammation. In addition, it is possible to suggest that via autocrine signalling, adiponectin amplifies the release of MCP-1 and PGE_2_ and COX-2 expression by preadipocytes under the stimulus of a metalloproteinase. To our knowledge, this is the first demonstration of the ability of a metalloproteinase to trigger adiponectin and MCP-1 synthesis by adipose tissue cells.

In conclusion, the data from the present study show, for the first time, the ability of a representative single-domain metalloproteinase, BmooMPα-I, to activate an inflammatory response in preadipocytes. BmooMPα-I induced the release of the inflammatory mediators PGE_2_, PGI_2_, MCP-1, and adiponectin but not IL-6, KC, leptin, or resistin by preadipocytes. BmooMPα-I-induced PGE_2_ biosynthesis was dependent on sPLA_2_-IIA, cPLA_2_-α, COX-1, and COX-2 pathways. Moreover, BmooMPα-I upregulated COX-2 protein expression but did not alter mPGES-1 expression. Furthermore, we demonstrated that BmooMPα-I enzymatic activity is essential for the activation of prostanoid biosynthesis in preadipocytes (Scheme 1). Altogether, our data highlight preadipocytes as important targets for the action of metalloproteinases and provide new insights into the contributions of these enzymes to the inflammation of adipose tissue and tissues adjacent to it. Furthermore, our study provides insight into the importance of the catalytic domain of MMPs to the inflammatory activity triggered by these enzymes.

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
