# Peer review of "A Metalloproteinase Induces an Inflammatory Response in Preadipocytes with the Activation of COX Signalling Pathways and Participation of Endogenous Phospholipases A2"

_biomolecules, 2021, doi:10.3390/biom11070921_

Round 1

Reviewer 1 Report

The findings are very interesting. The approach is elegant in that conclusions from studies on a venom metalloproteinase are extrapolated to propose roles for MMPs in the regulation of adipose tissue inflammation. The experimental data are very nicely presented.

This reviewer encourages the authors to choose the Special Issue of Biomolecules titled "Matrix Metalloproteinases in Health and Disease in the Times of COVID-19", as a venue for the revised version of this paper. This Special Issue would be an excellent venue as it has attracted a collection of excellent contributions on MMPs biology and this manuscript would be a nice addition, once it is thoroughly revised. 

However, this reviewer has identified the following major points that require improvement:

Introduction

While the regulation of prostaglandins by MMPs is of a great interest, several previous publications have addressed this phenomenon.  For instance: 

Identification of a Novel Heart-Liver Axis: Matrix Metalloproteinase-2 Negatively Regulates Cardiac Secreted Phospholipase A2 to Modulate Lipid Metabolism and Inflammation in the Liver.  J Am Heart Assoc. 2015 Nov 13;4(11):e002553. doi: 10.1161/JAHA.115.002553.

These contributions should be acknowledged and discussed.

Results

1) It is imperative that studies are conducted to include a concentration response analysis for BmooMPα-I effects. Please provide the molar concentration and enzymatic activity of BmooMPα-I. Please discuss  how the effective concentration(s) of BmooMPα-I compare(s) with the concentration of relevant mammalian MMPs either in tissue or circulation. Without these data it is impossible to ascertain whether the conclusions made for BmooMPα-I (at 6 microgram/mL...molar concentration???) can be extrapolated to MMPs (which are found at nanomolar concentrations in serum). This reviewer understands that there might be quantitative differences in concentration, enzymatic activity for BmooMPα-I, compared to relevant MMPs in the mammalian circulation. However, adding the above requested data would greatly improve this otherwise nicely executed and presented study. It would be great if the authors include a diagram with a hypothetical mechanism of action of BmooMPα-I addressing how such a mechanism would be elicited by MMPs.

2) The data suggest no effect effect of BmooMPα-I on COX 1, as opposed to COX2. Could the authors propose/discuss a plausible mechanism for these differences?

3)  The pharmacological inhibition of cPLA2, iPLA2 and sPLA2 is likely not sufficient for identification of a PLA2 class. Two of the inhibitors are used at 1 micromolar the third inhibitor is used at 10 micromolar. A concentration-response to these inhibitors should be presented. What is the mechanism of BmooMPα-I regulation of PLA2 activity? How does this mechanism translate to MMPs? Addressing these questions could be a nice addition to the earlier requested mechanistic figure.

Author Response

Dear Reviewer,

We appreciate the careful revision of our manuscript and acknowledge the comments made.

Please see the answers in the attachment.

Reviewer 2 Report

In this study, Janovits et al. want to investigate the inflammatory response in preadipocytes and in response to metalloproteinases. To do so, they stimulate cultured mouse 3T3-L1 cells with a metalloproteinase extracted form snake venom and show that this activates COX signaling pathways. The manuscript is well written and the data are clearly presented. However, I have concerns regarding the set-up of the study. Please find below my detailed comments.

  1. My main concern is the generalization that this metalloproteinase derived from snake venom represents the effect of metalloproteinases on adipocytes. This is not correct. Although metalloproteinase have high active site similarity, they do have different substrate repertoires and additional domains that alter their functionality. Therefore, the authors should not generalize this effect and specifically mention the metalloproteinase that they are studying.
  2. I wonder how sure the authors are of the purity of the isolated metalloproteinase? It is mentioned that the purity was evaluated by SDS-PAGE. However, what about other contaminants such as endotoxins? These could have a major effect on inflammatory pathways and their presence should be evaluated.
  3. Statistics: all experiments shown are based on 3-4 data-points. In such case, non-parametric statistical tests and descriptive statistics (median,…) should be used.
  4. How sure are the authors that the effect of batimastat is directly due to inhibition of the added metalloproteinase? It is known that Batimastat has general anti-inflammatory activity.
  5. Line 160 – please correct: [ref]

Author Response

(The authors gave the same response as above.)

Round 2

Reviewer 1 Report

Thank you for the detailed response to this reviewers' comments. In line 464, the authors indicate that the regulation of sPLA2 by MMP-2 is in contrast to their findings. In the view of this reviewer, this interpretation is imprecise. Rather, both the authors' findings and the citation to which they refer to in line 464 show that PLA2 activity is increased by pro-inflammatory settings such as the one they studied and the one created by lack of MMP-2. The authors are encouraged to make this minor clarification that centers the regulation of PLA2 activity on inflammatory stimuli with MMPs acting as modulators that can increase (as well as decrease) inflammation signaling both when expressed in excess and when under-expressed.  

Reviewer 2 Report

I would like to thank the authors for carefully addressing all my comments.

Author Response

Thank you very much for your review.